# Mitigation of Membrane Wetting by Applying a Low Temperature Membrane Distillation

**DOI:** 10.3390/membranes10070158

**Published:** 2020-07-21

**Authors:** Marek Gryta

**Affiliations:** Faculty of Chemical Technology and Engineering, West Pomeranian University of Technology in Szczecin, ul. Pułaskiego 10, 70-322 Szczecin, Poland; marek.gryta@zut.edu.pl; Tel.: +91-449-4682

**Keywords:** membrane distillation, scaling, desalination, membrane wetting

## Abstract

The formation of deposits on the membrane surface during membrane distillation is considered as one of the main reasons for membrane wetting. To assess the intensity of this phenomenon, long-term studies were performed comparing the membrane wettability with non-fouling feed (NaCl solutions) and feeds containing various foulants (lake and Baltic Sea water). The polypropylene membranes used were non-wetted by NaCl solutions during several hundred hours of water desalination. However, the occurrence of CaCO_3_ or other salt crystallization caused the membranes to be partially wetted, especially when periodical membrane cleaning was applied. The scaling intensity was significantly reduced by lowering the feed temperature from 353 to 315 K, which additionally resulted in the limitation of the degree of membrane wetting.

## 1. Introduction

The preservation of the gas phase in the membrane pores during membrane distillation (MD) process is a key requirement for its implementation. The MD process is used for desalting water and the separation of aqueous solutions; hence, the condition of non-wetting is obtained in most cases using membranes made from materials with hydrophobic properties [1,2,3,4].

Research on the MD process has been carried out for over 50 years [4,5], and despite several thousand works, no universal MD membrane has been fabricated that would be resistant to wetting by various solutions. However, the results obtained from these studies indicate that, likewise to pressure-driven membrane processes, membranes with different properties should be used in the MD process. This will also allow for the reduction of the wetting of MD membranes during the long-term operation of modules under industrial conditions.

The type of the membrane used depends on the feed composition and the processes occurring during MD (e.g., CaCO_3_ scaling) [2,4,5,6,7]. For this reason, several types of MD membranes can be distinguished that could be used for the separation: clean feed (CF), fouled feed (FF), scaled feed (SF), biological feed (BF), and surface-active feed (SAF). The latter type enables the management of process waters from oil and gas extraction. Unfortunately, the oils and surfactants present in these waters quickly wet the membranes [8,9]; hence, the industrial implementation of MD is very difficult in this case.

Limiting the wetting of MD membranes by organic compounds is a difficult challenge. Special cases of organic fouling are perfluoroalkyl substances, wherein adsorbing on the membrane surface causes its hydrophobization, which increases its resistance to wetting by water. However, in [10] it has been reported that due to surface diffusion inside the pores, these compounds penetrate from the feed to the distillate. This phenomenon decreases organic rejection and worsens the quality of the distillate, which is similar to effects when the pores are wetted by water. Rapid wetting of membranes causes the presence of surfactants in the feed, such as sodium dodecyl sulfate (SDS) [11]. Even at low SDS concentration (over 0.1 mmol/L), the membranes were wetted during a few hours of the MD process. In both cases, wetting the pores caused by organic fouling can be limited by using surface-modified membranes. This concept was presented in several works, where membranes with additional thin layers were covered [12,13,14,15] or nanoparticles to membrane structure were added [4,16]. Unfortunately, long-term studies with intensive fouling and scaling have most often been omitted in these works. MD research in which scaling layer would be repeatedly removed using periodical module rinsing is necessary to demonstrate the industrial suitability of new types of membranes.

For MD separation of solutions with high surface tension (about 70 mN/m) from which nothing precipitates (CF case) the membranes made of polymers such as polypropylene (PP) or polytetrafluoroethylene (PTFE) are successfully used [4,5,17,18]. In this case, it is possible to use the MD for the production of demi-water from steam condensates or for the concentration of diluted salt solutions [4,5,19]. However, during the MD process of natural waters (e.g., seawater) the deposits are formed on the membrane surface (FF case), which facilitate the wetting of membranes [4,17,19]. This phenomenon is accelerated by scaling (SF case) since crystallizing salts can also cause membrane degradation [20]. In such cases, in addition to choosing the type of membranes, the limitation of their wetting also requires the application of appropriate technological solutions and selection of MD process conditions [4,18].

The number of publications presenting the fabrication of novel membranes for the MD process has considerably increased in recent years [2,4,5,6,7,16]. The formation of membrane structures with small pore sizes (below 0.2 μm) allows for high liquid enter pressure (LEP) values to be obtained, which hinders the wetting of membranes [4]. This enables the use of thin MD membranes, and due to a smaller resistance of mass transfer through the membrane, a larger permeate flux can be obtained [21,22]. For example, the permeate flux values of 80–120 L/m^2^h were achieved for membranes with thickness equal to 10–20 µm [1,2]. However, in most of these publications the wetting resistance was tested using distilled water or NaCl solutions (CF case) [4]. Nevertheless, a layer of salt deposit on the membrane surface during the desalination of seawater or brines is already formed after dozens of minutes of the MD process. The salt crystals may penetrate into the interior of the membrane wall up to 20–50 µm [23,24,25,26]. Therefore, wetting of thin membrane caused by deposit formation on their surface and sometimes also inside the membrane pores is usually observed [21,27,28,29].

The scaling intensity is being attempted to be limited using modified membranes. However, it should be taken into account that even on the super hydrophobic surfaces there will be a sufficient number of active sites for heterogeneous crystallization [30,31]. Moreover, the organic compounds, which are present in natural water, undergo the adsorption on the hydrophobic surface, which facilitates the salt crystallization [32]. For this reason, when the precipitates (scales) are formed on the membrane surface during water desalination [31], it should not be expected that the application of even the best super hydrophobic membranes will allow the membrane wettability to be eliminated.

In the case of high scaling intensities, the membranes selected for MD should be significantly thicker than those used in the MD process, wherein the fouling/scaling proceeds to a small degree [6,25]. Unfortunately, the effects of many publications discussing the membrane morphology and the operating conditions of the MD process are mainly appropriate for the latter case [1,2,3,4,5,21,22,23,24,25]. The application of membranes with thicker walls (e.g., 400–1000 µm) allowed for the long-term exploitation of the MD module despite the scaling phenomenon [4,19,33]; however, increasing the membrane thickness substantially decreases the productivity and effectiveness of the MD process [21,22].

The occurrence of scaling results in the necessity to apply a periodical membrane cleaning and/or to limit a scaling intensity. The latter can be achieved by using additional processes eliminating the oversaturation states of solutes in the feed before they flow into the MD module [34]. Moreover, the removal of deposits from the membrane surface (especially when the deposits penetrate into the pores), e.g., by rinsing of MD module with acid solutions, can be one of the reasons inducing the membrane wetting [19]. In such a case, it is advantageous to select the parameters of MD process, resulting in the formation of deposit only on the membrane surface [2,19,29,30].

Results from several studies [1,7,18,22,31,35,36,37] demonstrated that the intensity of scaling can be substantially reduced by decreasing the feed temperature, particularly for temperatures below 333 K. Under such conditions, the formation of sediment on the surface of membranes (e.g., CaCO_3_) is very slowed down. This makes it difficult to assess the actual benefits, especially since most of this work presented research results carried out most often for a period not exceeding 50 h [4]. During this time, small amounts of sediments are formed on the surface of the membranes, which does not allow to assess the scaling effect on membrane wetting. Such information can be obtained during the study of MD modules over much longer periods, preferably 1000 or more hours [19,38].

The evaporation efficiency decreased only from 90% to 75% in the case of lowering the feed temperature from 353 to 313 K [26,27,31]; however, the permeate flux was decreased several times which limited such a process solution. For this reason, such a solution can be considered when it is possible to obtain a large membrane area of MD modules using inexpensive membranes. This type of membranes includes capillary polypropylene membranes. In the present work, using long-term studies, it was examined whether lowering the feed temperature significantly reduces the wetting of PP membranes during water desalination.

## 2. Wetting by Scaling

In many publications, novel MD membranes are presented the non-wettability of which was obtained by creating the appropriate surface properties. However, in the case wherein the salts crystallize on the membrane surface during the MD process, the protective layers could be damaged, and the feed will begin to penetrate the pores of the membranes.

When the feed is a diluted solution, scaling can occur, most often for two reasons, as schematically shown in Figure 1. The first results from the fact that the feed is heated, which disturbs the bicarbonate balance and leads to CaCO_3_ scaling. This effect is described by Equation (1) [39]:
(1)$$2{HCO}_{3}^{-}+{Ca}^{2+}=H_{2}O+{CO}_{2}+{CaCO}_{3}$$

Lowering the process temperature below 323 K allows this phenomenon to be significantly reduced [18,39]. The heating of the feed causes the crystals nuclei to already be formed in the heat exchanger and the scaling layer to be formed at the inlet to the MD module. On the existing sediment, subsequent crystals are formed as a result of heterogenic crystallization, which decreases the state of oversaturation in the feed at the evaporation surface (Figure 1A, C_S_). For this reason, most of the CaCO_3_ sediment is located on the surface of the membranes, which causes the wetting of the surface pores but limits the wetting of the pores located deep in the wall [19]. In this case, membrane scaling can be significantly reduced by conducting the carbonate crystallization in mesh crystallizers assembled just before the inlet to the MD module [34]. Among other proposed solutions limiting the scaling phenomenon is the addition of antiscalants to the feed [36]; however, the presence of antiscalants hinders the retentate management.

The second reason for scaling is the increase in the feed concentration (from C_in_ to C_F_) due to water evaporation (J) along the MD module and especially the high solute content at the boundary layer (C_1_, Figure 1B). The liquid in this layer is stationary and the transport of solutes from the evaporation surface to the feed bulk can only take place through diffusion (dC/dx). Diffusion is a relatively slow process; hence, with a rapid increase in solutes concentration, supersaturation and crystallization may occur. The rate of water evaporation depends on the vapour pressure [40]. Its value for 313 K is 7.4 kPa, and at 353 K, it amounts 47.3 kPa; hence, a permeate flux of about 5 times higher is achieved for 353 K. For this reason, lowering the feed temperature (flux decreasing) limits the increase in the solute concentration in the evaporation layer (C_1_), which in turn can limit the membrane scaling [18].

The permeate obtained in the MD process comprises only 2–4% of the feed volume flowing into the module. For this reason, the retentate should be recycled to a feed tank in order to achieve a higher coefficient of water recovery. Such a mode of the MD process also allows the consumption of energy utilized for the heating of the feed to be reduced by almost 50% [32]. A drawback of the retentate recycle is a progressive increase of solute concentration in the feed, which can enhance the intensity of deposit formation on the membrane surface [1,28,41]. MD studies of seawater and RO retentate desalination confirmed that when the state of supersaturation was achieved, an intensive scaling and the complete blockage of the evaporation surface quickly takes place [23,24]. In such cases, a combination of MD installation with crystallization is necessary in order to limit scaling [37,42]. In the integrated MD-crystallizer systems, lowering of the feed temperature allows for the magnitude of concentration polarization to be decreased, which helps to limit the scaling [37].

Scaling is particularly dangerous because growing salt crystals can build up into the membrane wall and, by increasing their size, also destroy the membranes mechanically [19,42]. Therefore, salt crystallization on the surface of membranes in addition to wetting the pores can also cause mechanical degradation of the surface layer (Figure 2). The surface of the crystals is hydrophilic; hence, this phenomenon facilitates wetting the pores inside the membrane wall.

In order to reduce fouling and increase the resistance of membranes to wetting, it is proposed to cover membranes by micropillars or other surface structures [4,43]. Membranes modified in this way allow the air layer at the membrane surface to be retained (Figure 3A), which reduces fouling and, as a result, reduces wetting of the membranes [14,43]. In this case, evaporation occurs on the surface of the trapped air layer, which protects the membrane against direct influence of the feed. However, it also increases the concentration of solutes between the pillars. If supersaturation is achieved, the resulting salt crystals can destroy the protective layer (Figure 3B). A protective gas layer at the membrane surface can also be obtained by using feed flow with air bubbles [13]. However, as shown on the feed/air surface, crystallization nuclei are formed; thus, this process solution can also cause the wetting of membranes [44]. For this reason, long-term MD studies, in which intensive scaling has taken place, are necessary to assess the industrial suitability of new membranes and process solutions.

## 3. Materials and Methods

### 3.1. MD Experimental Set-up

MD studies were performed with the application of two types of capillary membranes (made from polypropylene), which were commercially fabricated for microfiltration process. The parameters of the membranes and modules assembled from these membranes are presented in Table 1. The applied membranes have a similar internal diameter equal to 1.8 mm, and a wall thickness of about 0.4 mm. These membranes were also used in other MD studies, which showed their good wetting resistance and suitability for industrial applications, especially for the CF case [19,33]. The availability of commercial membranes that are resistant to wetting is the basis for the development of the MD process [9,15,38].

The MD process was carried out continuously (five days per week) using the experimental set-up presented in Figure 4. The volume of the feed tank was 5 L and a process temperature was fixed in the range of 313–353 ± 1 K using a Nűga temperature regulator (Germany). The feed water was continuously dosed into the feed tank and a constant feed volume (4.5 L) was maintained. Since the cover of the feed tank was not airtight, a collection of the MD permeate flux as well as the natural evaporation caused a reduction of the feed volume. A feed stream flowed through the lumen of the capillaries during the MD experiments. MD modules without external housing (submerged MD module) were applied. These modules were equipped with 3 (4-M5) hydrophobic capillary membranes with a working length of about 22 cm.

A design of the installation allowed for simultaneous examination of two MD modules placed in separate distillate tanks (volume 0.7 L). These tanks were assembled inside a cooling bath, which was cooled by tap water, and the distillate temperature was maintained at a level of 290–293 K. The temperatures were measured using thermometers with ±0.2 K accuracy. At the beginning of each measurement series, the distillate loops were refilled by distilled water (1–3 μS/cm). The obtained permeate flux was calculated on the basis of changes in the distillate volume over the period examined.

### 3.2. Feed Solutions

The wetting of the hydrophobic membrane during the MD process was influenced by a long-term contact with supplied water, a composition (salinity) of the feed, and the formation of sediments on the membrane surface. In order to investigate the effect of these factors on the wetting of the studied capillary PP membranes, NaCl solutions (non-scaling feed), lake water (CaCO_3_ scaling), and brackish water from the Baltic Sea were used as feed. Distilled water was also applied as a feed to evaluate the efficiency of the MD modules used (feed 313–353 K).

The pre-treated lake water (coagulation and filtration) was used as feeding water for MD studies with CaCO_3_ scaling. The electrical conductivity of this water was 625 ± 10 μS/cm, and the average concentrations of Si (ICP-AES analysis, Jobin Yvon).) was 2.7 ± 0.1 mg/L. The alkalinity analysis revealed that the HCO_3_^−^ concentration was equal to 2.3 ± 0.1 mmol/L. The MD studies were carried out for several weeks. The MD retentate was exchanged for a new portion of lake water in the feed tank once a week. The ion concentrations (IC analysis) in the feed water during the tested period are given in Table 2. The composition of the obtained MD retentate is also shown in this table.

Due to a complex composition and high concentration of salts, intensive scaling can occur during seawater desalination. Studies of water desalination were carried out using the water collected from the Baltic Sea. This sea is an inland water region located in Northern Europe, with an average depth of 52.3 m and area of more than 400,000 km^2^. The Baltic Sea is connected to the North Sea through the Danish Straits. The Baltic Sea is fed by more than 250 rivers that provide freshwater inflow. The relatively low annual mean temperatures of Northern Europe limit the intensity of evaporation, and the water salinity level evaluated was on an average 7 g/L; hence, this seawater is classified as brackish water.

A sample of the Baltic Sea water was collected in September (water temperature 17.8 °C) at the seashore in Dziwnów (Poland). In order to remove the suspended matter, the collected water was preliminary filtered through a paper filter (5 μm). The composition of these brackish water is presented in Table 2.

Laboratory scale desalination studies were started using the NaCl solution (7 g/L) as a feed, which allowed for the determination of the efficiency of MD modules and the intensity of membrane wetting during the MD process without membrane scaling.

### 3.3. Analytical Methods

Scanning electron microscopy (SEM) (SU-8000, Hitachi High Technologies Co., Tokyo, Japan) was employed for the determination of the membrane morphology. The composition of the deposits formed on the membrane surfaces was examined using the energy dispersion spectrometer method (EDS). After accomplishing each MD series, the membranes were rinsed with distilled water in order to remove the feed from the membrane surface. One membrane was collected from the module, and the holes formed after this membrane was glued (Figure 5). Such protection allowed for the testing of the efficiency of this module to continue, e.g., after flushing membranes with HCl solution. The membrane samples were dried at an ambient temperature prior to SEM examinations. The cross-section observations were performed with capillary membrane samples fractured in liquid nitrogen. All the samples were sputter coated with chromium using Q150T ES coater (Quorum Technologies Ltd., Lewes, UK) prior to examinations.

A mercury porosimetry method (Autopore III, Micrometrics GmbH, Aachen, Germany) was applied to determine the various parameters of the membranes structure.

An apparatus 6P Ultrameter (Myron L Company, Carlsbad, CA, USA) was employed to evaluate the electrical conductivity of tested solutions.

The composition and structure of the precipitated scale layer (crystals ICPDS database) were analyzed using an Empyrean PANalytical diffractometer (Panalytical Ltd., Almelo, The Nederland).

An ion chromatography method was utilized to measure the anion and cation concentrations. An 850 Professional IC chromatograph (Herisau Metrohm, Switherland) with a conductivity detector was applied for these studies. The anions were separated on a 1.7 × 3.5 mm Metrosep RP guard column in series with a 250 × 4.0 mm Metrohm A Supp5–250 analytical column. The aqueous eluent contained 3.2 mmol/L Na_2_CO_3_ + 1.0 mmol/L NaHCO_3_ (flow rate 0.7 mL/min). The cations analysis was performed with an A C2 Guard column in series with a 150 × 4.0 mm Metrosep C2-150 analytical column. In this case, the eluent was a mixture of tartaric acid (4 mmol/L) with 0.75 mmol/L 2-picoline acid.

The membrane hydrophobicity was determined using a Sigma 701 microbalance (KSV Instrument, Ltd., Espoo, Finland). The Wilhelmy plate method, allowing for the determination of the dynamic contact angle, was applied.

The change in membrane hydrophobicity caused by scaling was assessed by observing the effect of its surface interaction with the fluorescein dye (fluorescein sodium salt, Sigma-Aldrich Sp.z.o.o., Poznań, Poland). The dye drop, put on the membrane surface, was removed after 15 min by washing the membrane with distilled water, followed by fluorescence observations of the membrane surface using a laser scanning microscope (Zeiss LSM 510, Carl Zeiss, Jena, Germany). Dye adsorption occurred at the hydrophilic sites, resulting in an intense green glow when the membrane surface was observed through the LSM.

## 4. Results and Discussion

### 4.1. Membrane Performance

The membrane resistance for wetting is affected by both the surface morphology and the wall structure. Capillary membranes with the spongy structure were used in this study. SEM images of the membrane surface on the feed side are presented in Figure 6. The images of the surface pores are similar in both cases, despite the fact that these membranes were provided by two different manufacturers. The tested membranes also possess the similar values of the pore size and the porosity (Table 3).

A similarity in the membrane structures was also confirmed by the obtained results of the research on the influence of the feed temperature on the MD efficiency (Figure 7), which was carried out under the same hydrodynamic conditions. The properties’ similarity of the used membrane enabled a comparison of the effect of scaling on their wetting. This is difficult if we are testing membranes with different properties. In such a case, the differences in wetting are rather due to the different intensity of scaling caused by differences in solute concentration at the boundary layer due to differences in the permeate flux.

The driving force of mass transfer in the MD process is a difference in the vapour pressure, which increases exponentially along with the increase of the feed temperature (T_F_) [40], and this exponential character is reflected in the changes in the permeate flux presented in Figure 7. The obtained process efficiency for the feed temperature in the range of 343–353 K was more than 5 times higher than that obtained for T_F_ = 313–323 K. With regard to this, there should be a serious reason to utilize a low temperature for the operation of the MD process. One of such reasons is the limitation of membrane wetting caused by scaling and in order to avoid an irreversible damage of MD modules [18,23,33,37].

The evaporation of water and heat losses reduced the membrane temperature on the feed side from T_F_ to T_1_ (Figure 1A) which resulted in a decrease of the permeate flux [40]. The module efficiency can be enhanced by increasing the turbulence of the feed flow. However, it is worthy to note that for low temperatures of the MD process (e.g., below 323 K), the improvement of conditions of heat exchange did not increase the MD process yield, but only led to the enhancement of heat losses through the conductivity [45].

### 4.2. The Effect of CaCO_3_ Scaling

It was demonstrated that an immersion of hydrophobic membrane in distilled water (or diluted solution) for a longer period may cause the changes in the membrane properties [19,33,46]. Therefore, it is difficult to evaluate the effect of scaling on wettability without the knowledge of membrane wetting only by clean water. In order to demonstrate the intensity of these phenomena, the measurement series were carried out in the applied MD system using distilled water and lake water as a feed.

The productivity of the MD process (Accurel PP S6/2 membranes) was closed to 30 L/m^2^h at the temperature of 353 K (Figure 7 and Figure 8). However, the permeate flux was quickly decreased when the lake water was applied (Figure 8, M2), because such a high feed temperature causes intensive scaling, as was demonstrated in previous works [4,5,19]. A deposit formed on the membrane surface accelerates the membrane wetting [4]; therefore, long-term studies should be performed to evaluate membrane performance [33,46].

A long-term study of the MD process (over one month) was performed in the first stage using the distilled water with NaCl (1 g/L) as a feed. Such a diluted solution does not cause scaling, and simultaneously allows the appearance of possible leakages of the feed through wetted pores of the membrane to be detected. The experimental results were presented in Figure 8 (M1 module). The initial yield of the MD process amounted to 29.7 L/m^2^h. A stable permeate flux and over 99.9% degree of salt retention (feed 1950 μS/cm) was achieved during 650 h of the MD process. After this period, the permeate flux was equal to 28.5 L/m^2^h and the distillate electrical conductivity was at a level of 1.42 μS/cm. Such results confirmed the conclusions from previous works [19,21,34,46]: That the Accurel PP S6/2 membranes used demonstrate a good resistance for wettability and can be applied in the MD process.

In the second series (M2 module) lake water containing a significant amount of HCO_3_^−^ ions (2.3 ± 0.1 mmol/L) was used. A permeate flux equal to 29.1 L/m^2^h was maintained for the initial 60 h of the MD process, and then the module efficiency started to systematically decrease. The final permeate flux amounted to 17.5 L/m^2^h after 670 h of process operation, hence, by 40% lower than the initial value. The SEM examination of membrane samples confirmed that the main reason of such a decline of module efficiency was the formation of crystalline deposit on the membrane surface (Figure 9). The results of SEM-EDS analysis indicated that the Ca and O were the major components of these crystals, which indicated the presence of CaCO_3_. The results of XRD analysis indicated that the calcite crystals were formed (Table 4, 353 K). The large rhombic crystals, characteristic of calcite [47], covered almost all the membrane surface. The SEM examination of the cross-section revealed that the deposit was mainly formed on the membrane surface and the crystals do not penetrate into the pore interior (Figure 9B).

The SEM examination revealed that several new crystals were formed on the surface of calcite crystals due to the secondary nucleation, which caused a deposit layer to be mainly accumulated over the membrane surface. The crystal layer deposited on the membrane surface had a porous structure (Figure 9B), and the slits between the crystals enabled the feed to flow to the membrane. However, a thickness of the formed deposit amounted to 10–30 μm; hence, a significant resistance for heat transfer was created. As a result, a temperature of the evaporation surface was lowered [33,40], which was one of the main reasons causing the observed decline of the MD process efficiency (Figure 8, M2).

The initial efficiency of the M2 module was restored by rinsing the membranes with 3 wt% solution of HCl, although an increase in the distillate conductivity was also observed (Figure 8, HCl). This fact confirms the results of other works, reporting that a CaCO_3_ deposit is easily removable when the membranes are rinsed with acid solutions; however, rinsing repetition significantly accelerates membrane wetting [19,33,48]. The deposit covering the membrane surface sometimes penetrates into the membrane pores (Figure 2B), which accelerates their wettability when the scales are replaced by the HCl solution. The examination of the contact angle also confirmed the partial wetting of the membrane surface. The measured contact angle value for the new Accurel PP S6/2 membranes amounted to 102° (advancing) and was decreased to 65° for membrane samples collected from the M2 module after their rinsing with a 3 wt% HCl solution.

The scaling effect on the wetting of membrane surfaces was confirmed by using fluorescent dye in LSM observations. Fluorescein practically did not bind to the hydrophobic surface of the new membrane (Figure 10A). This situation changed when the CaCO_3_ precipitate formed on the membrane surface was removed by rinsing the module with HCl solution. The numerous intensely luminous spots visible during LSM examination (Figure 10B) were due to dye adsorption, which indicated the hydrophilization of these fragments of membrane surface.

The SEM images of the membrane sample taken from the M2 module after rinsing by acid solution were close to that presented in Figure 6A (new membrane). However, MD studies presented in work [33] have shown that repetition of the process: CaCO_3_ crystallization—HCl rinsing causes degradation of the membrane surface (Figure 11). This degradation degree was obtained by using module rinsing 7 times with HCl solution during over 750 h of the MD process duration.

### 4.3. Low Temperature MD

The results presented in previous paragraphs confirm how essential the elaboration of the conditions which eliminate the formation of deposits on the surface of MD membranes during the water desalination is. The results of a few works indicate that the formation of deposits can be limited by decreasing the feed temperature to a level of 303–323 K [18,27,35,37]. However, in these works, MD research was conducted for a short period, below 50 h, which makes it difficult to evaluate this solution for industrial installations. In previous work [46], it was revealed that a low feed temperature does not completely eliminate scaling. Therefore, during many months of the MD process, the progressive accumulation of the scaling layer on the membrane surface may also cause membrane wetting. Hence, further studies on the effectiveness of low temperature MD are required.

In order to check this MD process solution, long-term studies were carried out feeding the M3 module with lake water at a temperature of 315 K. For this feed temperature, a stable flux at a level of 3.2 L/m^2^h was achieved (Figure 8, first 300 h). During the initial 200 h period, the value of the distillate electrical conductivity was stabilized at a level of 2.3 μS/cm, which confirmed that the membrane pores were non-wetted. However, in the following hours of M3 module exploitation, a pronounced decline of module efficiency and a slight increase of distillate electrical conductivity was observed (Figure 8, from 300 h). Finally, the permeate flux decreased from 3.2 to 2.6 L/m^2^h and the distillate conductivity increased from 2.3 to 4.4 μS/cm during 670h of the MD process operation (Figure 8, M3). The results from these relationships show that the application of lower feed temperature could significantly diminish the intensity of flux decline, but did not prevent a deterioration of the membrane transport properties. However, the SEM examinations (Figure 12) revealed that the reasons for this were different than those found during the application of a higher feed temperature (353 K).

In accordance with the expectation [18,46], lowering the feed temperature almost allowed for elimination of scaling. The presence of deposit was found only in a relatively few places on the membrane surface (Figure 12A). The large calcite crystals (10–30 μm) shown in Figure 9 were not observed; only the amorphous deposits and small crystals with a dimension below 0.5 μm were formed (Figure 12B). The thickness of this deposit layer was below 1 μm (Figure 12C). The SEM-EDS analysis indicated that the formed deposit mostly contained Ca, O, and very small amount of Mg and Si. Although the SEM images were very different (Figure 9 and Figure 12), the results of the XRD analysis were similar (Figure 13), because in both cases the formed deposits contained mainly calcite. However, the study showed that beside calcite, two types of magnesium calcium carbonate were also present in the sediment formed at low temperature (Table 4). It is noteworthy that the detected calcite (code 04-002-9082) had a slightly smaller unit cell size compared to the calcite created in 353 K (code 04-012-0489).

During the low temperature MD a significant fraction of deposits was formed on the edges of the inlet to the surface pores (Figure 12B), which could facilitate the inflow of the feed into their interior. With regard to this, it can be assumed that the wetting of the surface pores was a reason for the observed permeate flux decline, which was also confirmed by the noticed increase of distillate electrical conductivity (Figure 8, M3). However, the permeate flux was several times lower for 315 K than that obtained for 353 K; thus, due to a similar level of distillate conductivity and assuming a similar flux of salt transferred from the feed into the distillate, it can be stated that at 353 K the membranes underwent the wettability to a larger degree.

The above described results of investigations confirmed how for the evaluation of membrane properties and the effect of process conditions it is necessary to carry out long-term MD studies. However, these investigations were conducted with the feed containing a small amount of salt, such as lake water (Table 2). With regard to this, in the next part of the work, a long-term MD study of water from the Baltic Sea was undertaken to confirm the advantages of running the process with the utilization of low feed temperature.

### 4.4. Water Desalination

It is worth noting that the processes causing membrane wetting can proceed very slowly. Therefore, before the studies on the influence of scaling, distilled water with the addition of NaCl (7 g/L) was used for the MD experiments in order to stabilize the separation properties of applied membranes. This NaCl solution with a salt concentration close to that of the water from the Baltic Sea did not contain the components that could cause scaling. Therefore, it allowed us to study whether the fact of membrane long-term contact itself with pure salt solution has a significant impact on membrane pore wetting.

In accordance with the data presented in Figure 7, lowering the feed temperature to 315 K significantly decreased the efficiency of MD process. A similar permeate flux at a level of 3.1 L/m^2^h was achieved for the membranes Accurel PP S6/2 and C-PP (Figure 14). The MD process was carried out for almost 500 h and the productivity of the tested MD modules did not undergo changes over this period, and the distillate conductivity had low values. Such results confirmed that the applied PP membranes maintained their non-wettability during the MD process of clean NaCl solution.

The results presented in Figure 14 proved that in the initial period of exploitation of MD modules, both permeate flux and distillate conductivity membranes did not deteriorate with the process time, which indicated a lack of wetting of membranes. Some changes in these parameters were noticed until after 100 h of the process. In the case of membrane PVDF, an increase in its wetting intensity occurred after 300–500 h of modules operation [49]. This result confirms that the assessment of membrane resistance to wetting requires MD testing for at least several hundred hours.

Although the values of the electrical conductivity of the obtained distillate were slightly increased (Figure 14) due to the wetting of certain pores, the distillate conductivity was then stabilized, which confirmed the necessity of preliminary stabilization. As demonstrated in the previous works [19,33], this results from the fact that the polypropylene membranes undergo surface wettability during the initial period of MD process. Moreover, the changes observed in the values of electrical conductivity were different for the particular membranes, which indicated their different resistance for the surface wettability. The distillate conductivity started to increase most rapidly in the case of C-PP membranes after already 150 h, whereas in the case of S6/2 membranes, the same trend was observed only after 250 h of MD process operation. However, the increase of conductivity had a short-time character and after about 300 h of MD, a value of the distillate conductivity obtained from modules M4 and M5 was stabilized at a level of 4.5 and 8.1 μS/cm, respectively.

It is also worth noticing that the feed contained 7 g/L of salt and obtained such low conductivity values (Figure 14) indicated that only a relatively few pores were wetted. Moreover, the occurrence of an increase in the distillate conductivity during the initial period of module operation is not always associated with increasing wetting of membranes. During the membrane fabrication in their structure, certain defects may be created locally, such as pores with a large diameter or accidental contaminations, which facilitate the pore wetting. With regard to this, it should be expected that during the initial period of MD module exploitation, a small fraction of the pores would become wetted. Taking into consideration this fact, it is essential to apply the preliminary stabilization, e.g., using a non-scaling feed in order to evaluate the changes of the membrane properties in the MD process.

In our case, after the initial 300 h of MD studies, the values of electrical conductivity were not changed during a consecutive 200 h of the process (Figure 14), which indicates that the wetting phenomenon of successive membrane pores was limited. Hence, it was assumed that in the case of the replacement of NaCl solution by Baltic Sea water, the observation of the occurrence of a future increase of conductivity would result from the pore wetting accelerated by membrane scaling.

The study of the separation of water from the Baltic Sea was carried out for almost 600 h; therefore, the total exploitation time testing the M5 module amounted to 1100 h (Figure 15). The feed concentration increased over 5 times (Figure 16), which corresponds to an 80% degree of water recovery due to a continuous supplementation of water in the feed tank. Despite such a significant increase of salt concentration in the feed, the obtained module efficiency was stable and close to the values obtained during the separation of NaCl solution (Figure 14). This fact confirms another important advantage of MD process, namely, that the increase of the feed concentration has a slight influence on the process efficiency, as reported many times in the literature [4,18,25,28,33].

Despite the significant increase in the feed concentration, the distillate conductivity increased slightly above 10 μS/cm (Figure 15). This confirms the fact that only a relatively small part of the membrane walls was wetted during 1100 h of the M5 module exploitations. Moreover, the fact that the permeate flux decreased in a small degree during 600 h desalination of the Baltic Sea water also indicates that the C-PP membranes were not wetted, similarly as in the case of NaCl solution (Figure 14).

The SEM examinations confirmed that this was mainly due to the fact that only a relatively small amount of deposits was formed on the surface of the membranes (Figure 17). The intensity of scaling resulted from the feed concentration. After 1000 h of the process, dosing of Baltic Sea water to the feed tank was suspended and the volume of the feed was systematically reduced, which accelerated the increase of the salt concentration in the feed (Figure 16). IC studies showed that the content of Ca^2+^, SO_4_^2−^, and Mg^2+^ decreased in the feed after 1050 h of the process (Figure 16); hence, these components may form scaling. At the same time, a slight decrease in efficiency and an increase in the distillate conductivity were found (Figure 15, from 1000 h). These results indicated that for the conditions of the MD process used to maintain membrane non-wetting, the degree of concentration of Baltic Sea water should not exceed 80%, which was reached after about 500 h of the MD process.

### 4.5. MD Process of Brine

Lowering the feed temperature limits the rate of formation of the oversaturated state at the evaporation surface, which can affect both the intensity of scaling and the structure of formed deposits, as was confirmed by XRD studies (Table 4). To examine the effect of feed temperature lowering, the MD retentate obtained in the previous stage of study was used for further MD research. Dosing of the Baltic Sea water to the feed tank was continued, obtaining a systematic increase in the concentration of ions in the feed (Figure 18). The MD tests were carried out with the M5B module (the M5 module from which one capillary was removed) and the M4 module (Table 1), which was previously used for 500 h for the separation of NaCl solution (Figure 14).

The MD installation with MD retentate was operated for a period of almost 2 months. After this period, the feed concentration increased over two times, which allowed us to obtain brine containing over 100 g/L of salt. Supplementation of the Baltic Sea water to the feed tank caused the changes in the linear course of ion concentration. Moreover, a significant decrease in SO_4_^2−^ concentration was noticed after 400 h of the process, when their concentration approached 5 g/L (Figure 18, from 1500 h). In the case of the MD process of surface water, a sulphate precipitation in the MD process can take place after exceeding 1 g SO_4_^2−^/L, but a high concentration of NaCl significantly increases the sulphate solubility [50]. In the present case, a further increase in the NaCl content led to an increase in SO_4_^2−^ ions concentration to 5.5 g/L (Figure 18). Despite such a significant increase in the feed concentration and the occurrence of scaling, a surprisingly stable permeate flux was obtained. During the 600 h of MD process, the permeate flux decreased from the level of 3.4 to 3.2 L/m^2^h (Figure 19). A slightly lower efficiency was obtained for the M5B module, but the membranes assembled in this module were explored for 600 h longer (Figure 15), which could lead to their partial wetting. This is indicated by obtaining slightly higher distillate conductivity values that were stabilized at a value of 20 μS/cm (Figure 19, M5B module), while the conductivity of the distillate obtained from the M4 module was at a level of 10 μS/cm.

A relatively higher decline of the permeate flux was observed after modules rinsing with 3 wt% HCl (Figure 19, from 1700 h). This probably results from the fact that the rinsing operation caused the wettability of the surface pores that were previously covered by deposits due to slight scaling. The dissolution of deposits from the membrane surface increased the concentration of solutes, which diffuses through the wetted pores, resulted in a significant increase in the distillate conductivity (Figure 19, 1720 h). In the consecutive hours of the process, the obtained permeate diluted the water collected in the distillate tank and the conductivity of water recycled on the distillate side was decreased. However, its value was stabilized at a higher level than before rinsing the modules, which indicates that this operation increased the number of wetted pores. However, the obtained distillate conductivity values in both cases were very low and indicated almost 100% separation of solutes despite such a high salt concentration in the feed (about 100 g/L).

After the rinsing operation, the MD process of brine concentration was conducted for another 100 h (Figure 19). When the MD process was completed, the modules were rinsed with distilled water and the membrane samples were collected for SEM tests. The SEM examinations confirmed the formation of significant amounts of deposits on the membrane surface (Figure 20). Although the deposit covers most of the membrane surfaces, an area free from the deposits could be also observed. The amount of deposits and their structures were similar in both cases (Figure 20A,B). This indicated that it was not the type of membrane, but the process conditions and the feed concentration that determined the scaling process. The membrane cross-section images demonstrate that the thickness of the formed deposit amounted to 1–3 μm (Figure 20C,D). The deposits were bounded with the surface pores, but a strong tendency to their formation within the membrane pores was rather not observed.

The precipitates formed on the membrane surface crystallized in the form of rods, and an enlarged image of the local crystals is shown in Figure 20E. The composition of the deposits can be determined using SEM-EDS in a variant of the surface analyses. They revealed the dominant presence of Ca with a much lower intensity of the Cl peak. Despite many tests, no sulfur was found.

## 5. Conclusions

Long-term studies of the MD process have confirmed that determining the stability of the tested membranes and their resistance to wetting requires the MD tests to be conducted for at least several hundred hours.

The presence of deposits on the membrane surface and within the pores accelerated wetting of membranes. Dissolving sediments during module rinsing operation facilitated the penetration of solutions into the surface pores. Therefore, a cyclic repetition of the module rinsing could cause a significant increase in the degree of membrane wetting.

A lowering of the feed temperature from 353 to 315 K allowed us to significantly limit the intensity of CaCO_3_ scaling, and as a result, the MD process could be operated for several weeks without the necessity to clean the modules.

The SEM examinations revealed that lowering of the feed (lake water) temperature from 353 to 313 K significantly decreased the amount of scales and additionally changed the form of the deposits that were mainly formed by very small crystals of calcite and calcite magnesian. These deposits exhibited a low affinity to penetrate into the pore interior. In the case of the feed at a temperature of 353 K, the crystalline deposit strongly bounded with the membrane that was formed, which facilitated pore wetting during the module rinsing operation.

The SEM observations of membrane cross-sections confirmed that at a lower feed temperature, the deposit was mainly formed on the membrane surface. This was advantageous because it limited the pore wettability taking place during the removal of the deposit (e.g., by rinsing the module with HCl solutions).

During the MD process of Baltic Sea water (T_F_ = 315 K), the formation of scales was not found after reaching a coefficient of water recovery of about 80%. This resulted in a stable permeate flux without wetting the membrane. After achieving a 90% recovery of water, the amount of the deposits forming on the membrane surface increased significantly, but a substantial decline of the obtained permeate flux was still not observed. This result indicates that a lowering of the feed temperature to 315 K even allowed such a high coefficient of water recovery to be obtained during the MD process. However, a slight increase of distillate conductivity was observed after the module rinsing. This confirms that with regard to the possibility of deposit formation, the application of such high coefficients of water recovery can be a reason for the accelerated wetting of the membrane.

## Figures and Tables

**Figure 1 membranes-10-00158-f001:**
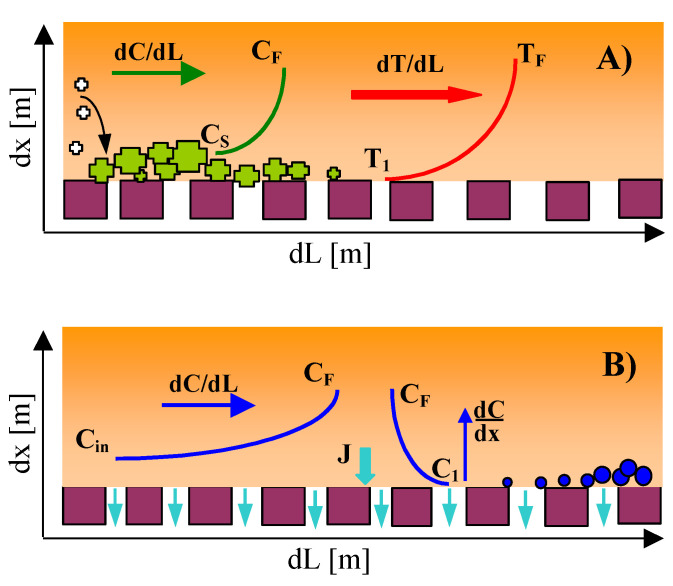
The changes in the solute concentration and liquid temperature on the feed side. (**A**) Oversaturated solution flows into the module, (**B**) oversaturation state created inside the membrane distillation (MD) module.

**Figure 2 membranes-10-00158-f002:**
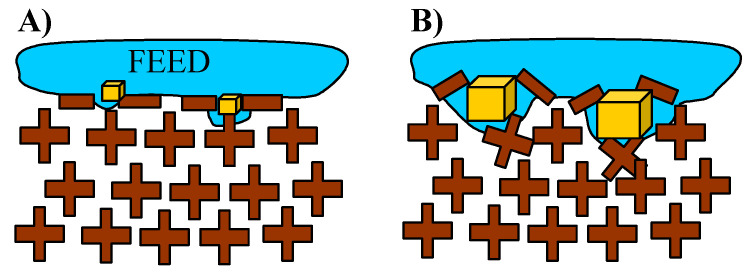
Mechanical degradation of pore structure caused by salt crystallization on the membrane surface. (**A**) scaling initiation (**B**) pore destruction by salt crystals.

**Figure 3 membranes-10-00158-f003:**
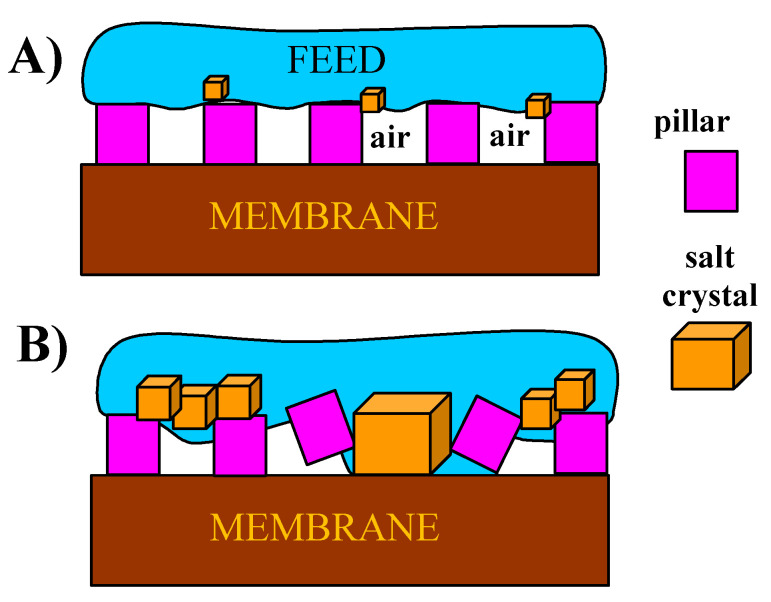
Mechanical destruction of pillar layers caused by salt crystallization. (**A**) scaling initiation (**B**) pillars destruction by salt crystals.

**Figure 4 membranes-10-00158-f004:**
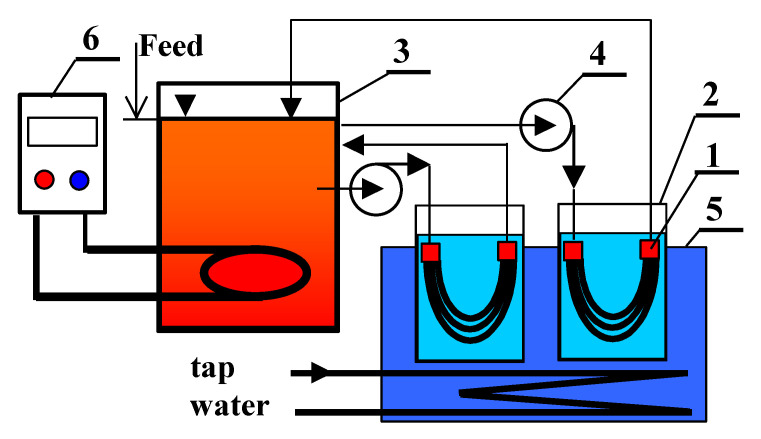
Experimental MD set-up. 1—submerged MD module, 2—distillate tank, 3—feed tank, 4—peristaltic pump, 5—cooling bath, 6—feed temperature regulator.

**Figure 5 membranes-10-00158-f005:**
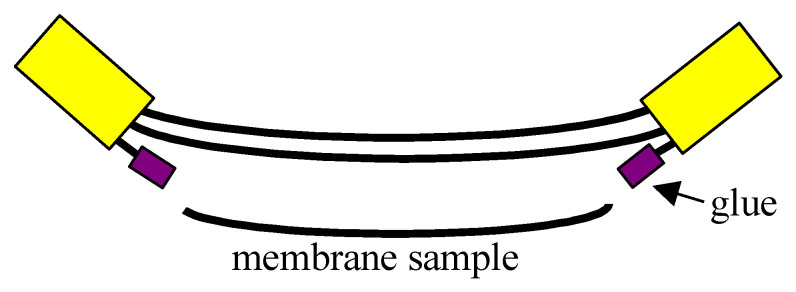
The way of membrane sample collection from submerged MD module.

**Figure 6 membranes-10-00158-f006:**
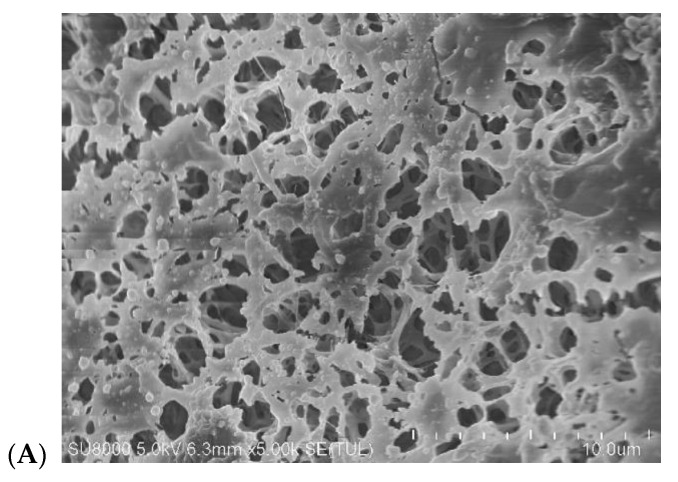
SEM images of capillary membrane surfaces on the lumen (feed) side. (**A**) Membrane Accurel polypropylene (PP) S6/2, (**B**) membrane C-PP.

**Figure 7 membranes-10-00158-f007:**
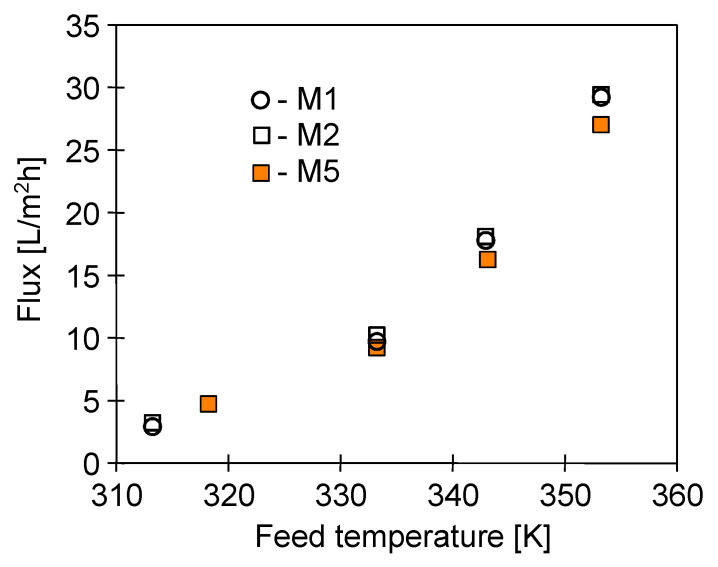
The influence of feed temperature on the permeate flux. Modules M1, M2, and M5. Feed flow velocity 0.6 m/s. Distillate temperature 291–293 K. Feed: Distilled water.

**Figure 8 membranes-10-00158-f008:**
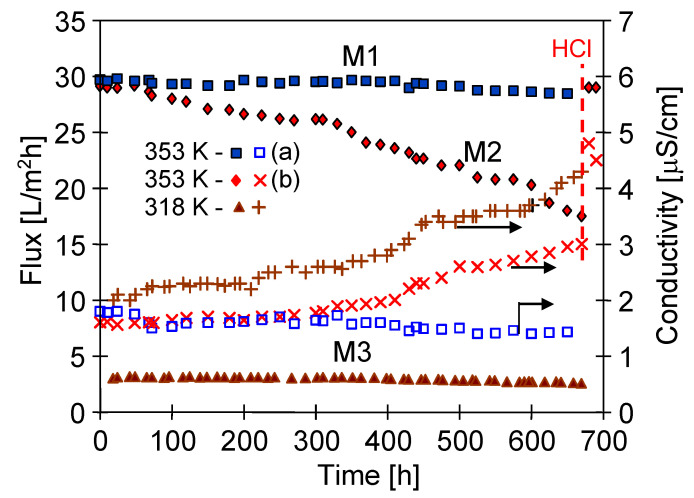
The influence of feed temperature on the changes in the permeate flux and distillate electrical conductivity. Feed: NaCl solution (1 g/L) and lake water (315 K and 353 K-b). Modules M1–M3 (membranes S6/2). HCl—module M2 rinsing with 3 wt% HCl solution.

**Figure 9 membranes-10-00158-f009:**
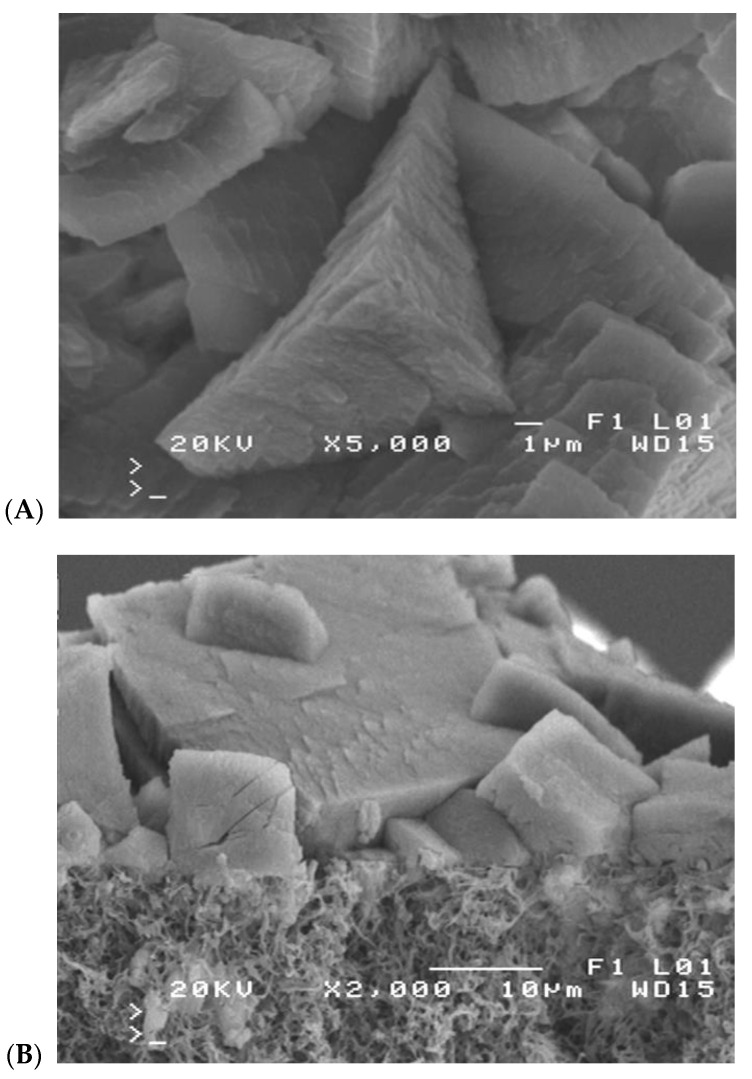
SEM images of CaCO_3_ deposit formed on the membrane surface at feed temperature 353 K. (**A**) Membrane surface covered by deposit, (**B**) membrane cross-section.

**Figure 10 membranes-10-00158-f010:**
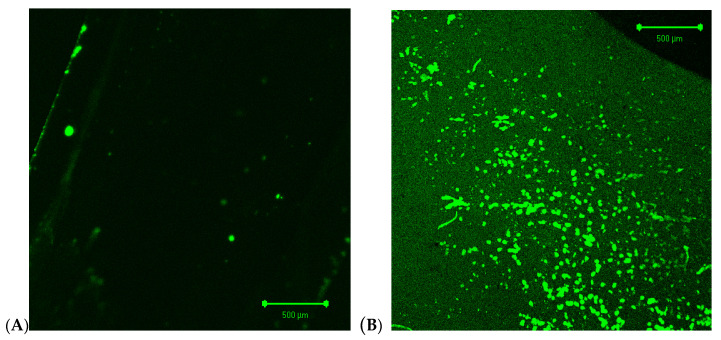
LSM images of Accurel PP S6/2 membrane surface covered by fluorescein. (**A**) New membrane, (**B**) membrane surface after removal of CaCO_3_ deposit by HCl solution.

**Figure 11 membranes-10-00158-f011:**
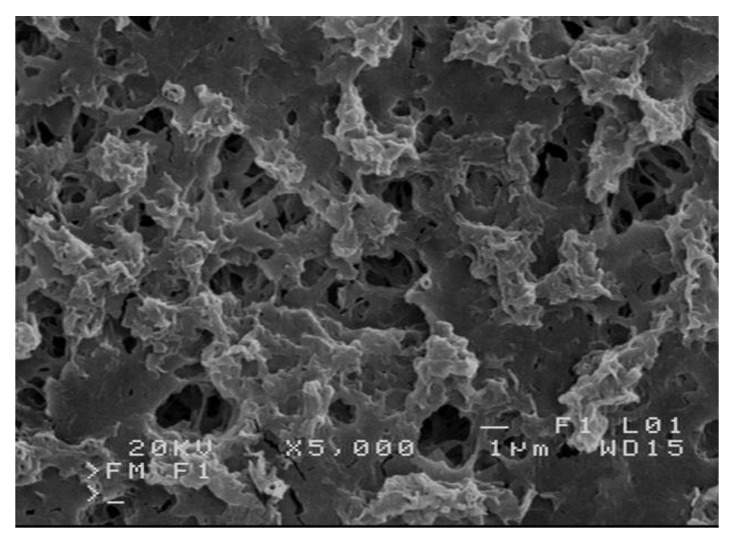
SEM images of membrane surface degraded by scaling and module rinsing by HCl solution (7 times) applied aiming to remove CaCO_3_ deposit formed during over 750 h of water desalination (T_F_ = 353 K).

**Figure 12 membranes-10-00158-f012:**
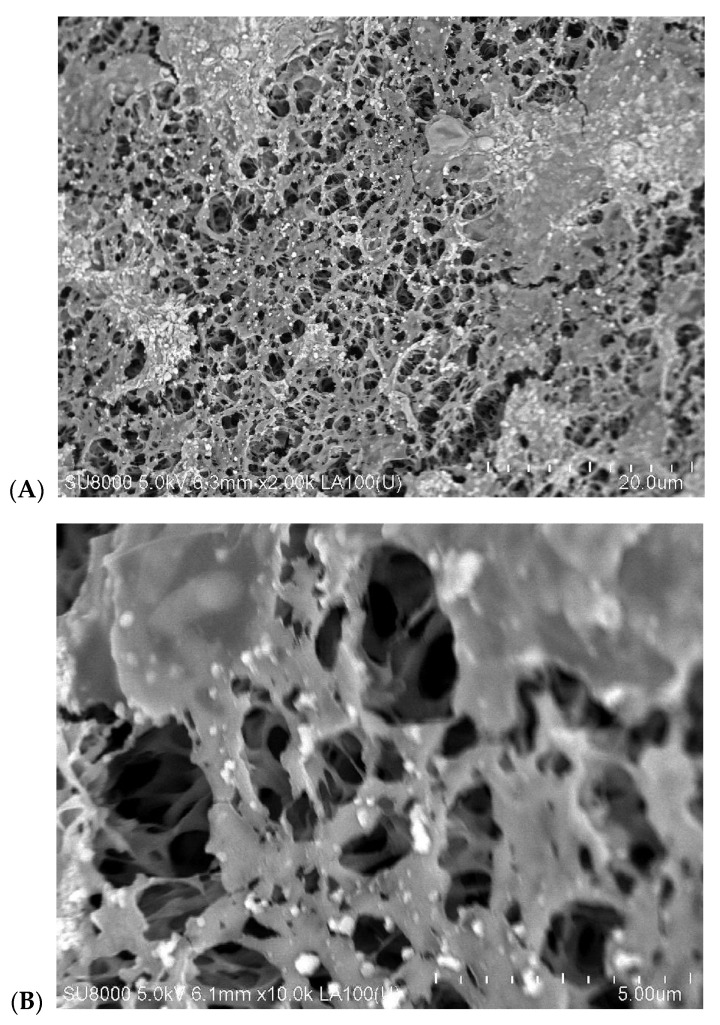
SEM images of deposit formed on the membrane surface (M3 module) at feed temperature 315 K. (**A**,**B**) membrane surface with deposit, (**C**) membrane cross-section.

**Figure 13 membranes-10-00158-f013:**
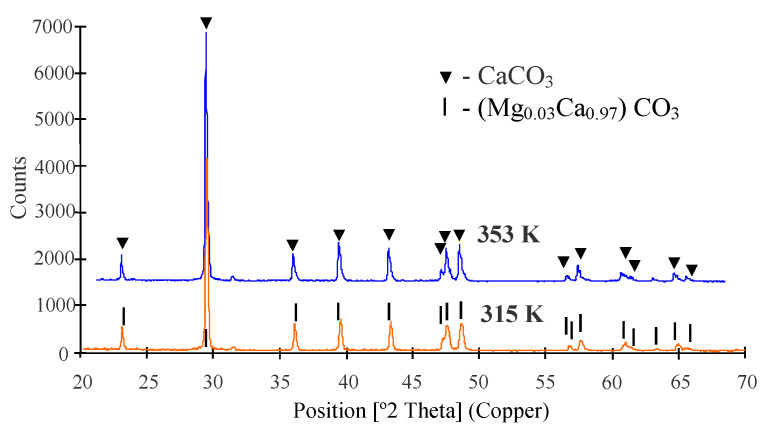
The results of XRD analysis.

**Figure 14 membranes-10-00158-f014:**
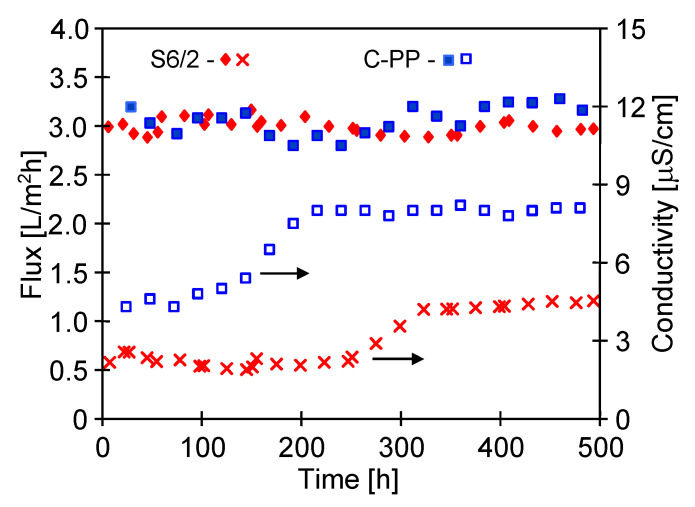
Changes in the permeate flux and distillate electrical conductivity during MD process of NaCl solution (initial concentration 7 g/L). Modules M4 (S6/2) and M5 (C-PP).

**Figure 15 membranes-10-00158-f015:**
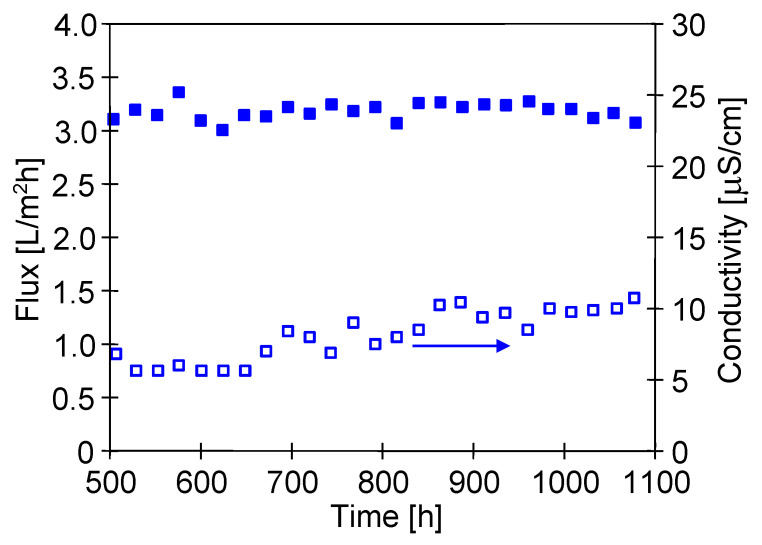
Changes in the permeate flux and distillate conductivity during desalination of Baltic Sea water. T_F_ = 315 K. Module M5.

**Figure 16 membranes-10-00158-f016:**
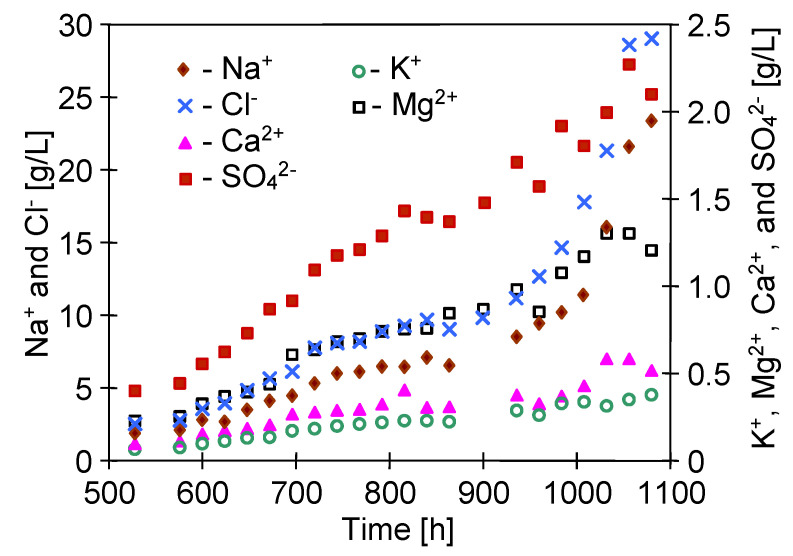
The changes in ion concentrations in the feed during desalination of Baltic Sea water.

**Figure 17 membranes-10-00158-f017:**
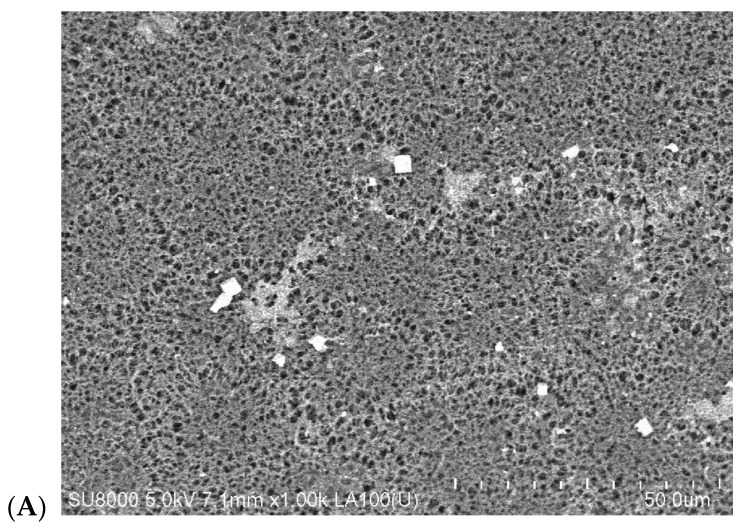
SEM images of membrane sample collected from M5 module after 600 h desalination of Baltic Sea water. Feed temperature 315 K. (**A**) Membrane surface with deposit, (**B**) membrane cross-section.

**Figure 18 membranes-10-00158-f018:**
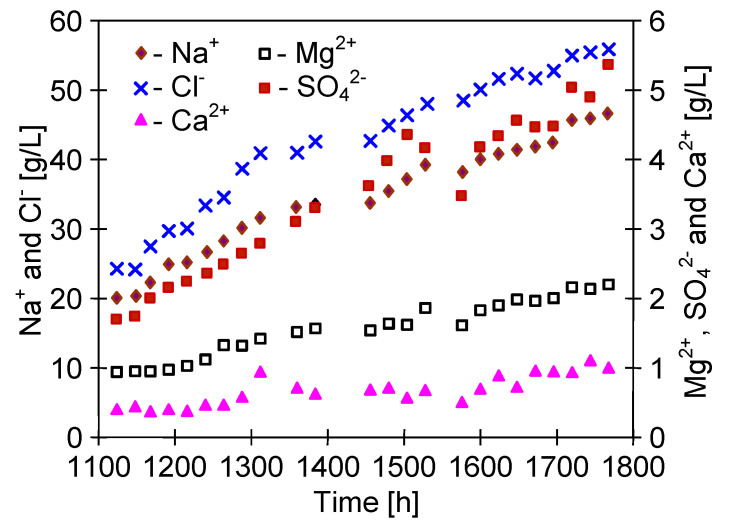
The changes of ion concentration in the feed during MD concentration of brine obtained from Baltic Sea water.

**Figure 19 membranes-10-00158-f019:**
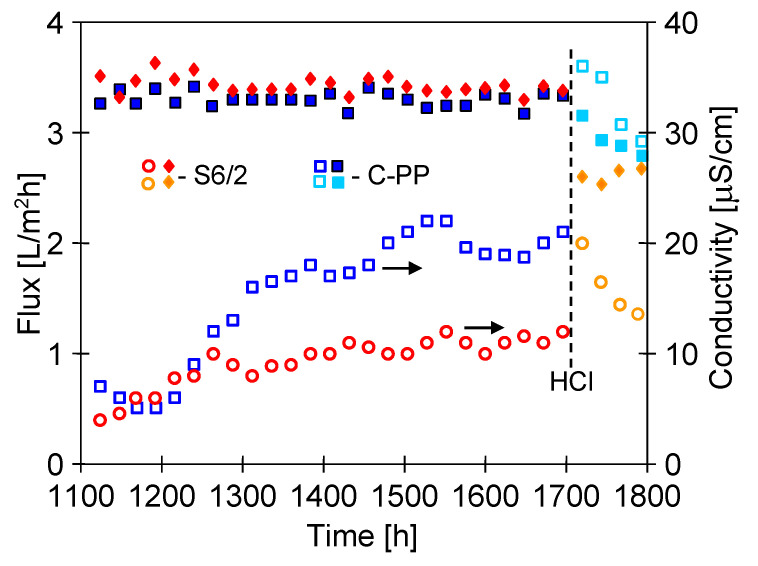
Changes of the permeate flux and distillate conductivity during MD process of brine obtained from of Baltic Sea water. Module M5B (membranes C-PP) and module M4 (membranes S6/2). HCl—modules rinsed with 3 wt% HCl.

**Figure 20 membranes-10-00158-f020:**
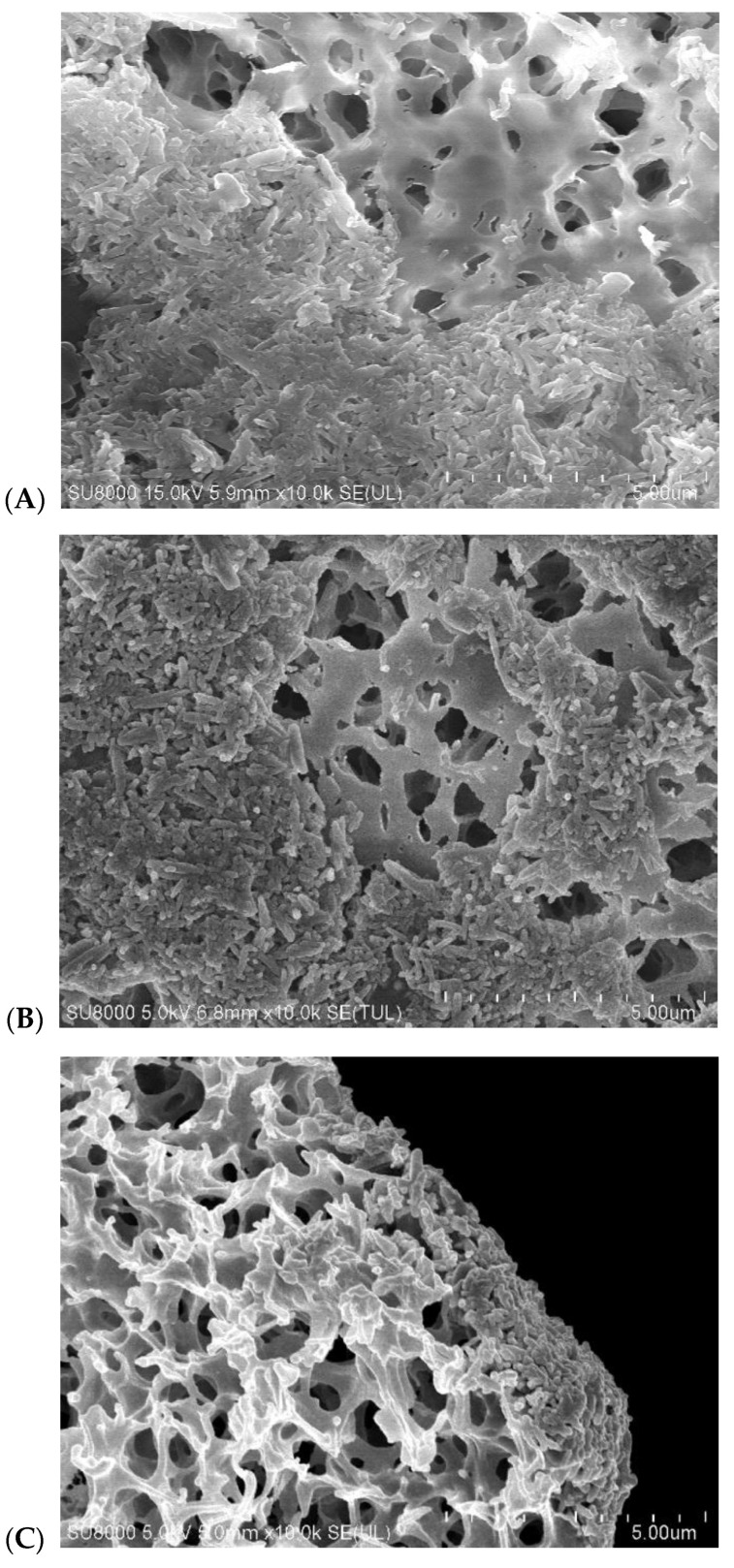
SEM images of deposits formed on the membrane surfaces during MD concentration of brine obtained from Baltic Sea water. Membrane Accurel PP S6/2—(**A**,**C**), membrane C-PP—(**B**,**D**,**E**). Images (**C**,**D**)—membrane cross-section with deposit.

**Table 1 membranes-10-00158-t001:** Parameters of the membranes and MD submerged modules employed.

Manufacturer	Membrane	Contact Angle	Module	Length (cm)	Number of Capillaries	Area (cm^2^)
Membrana GmbH Germany	Accurel PP S6/2	98	M1 M2 M3 M4	22.0 21,8 22.3 22.0	3 3 3 3	37.3 37.0 37.8 37.3
Polymemtech Poland	C-PP	96	M5 M5B	22.0 22.0	4 3	49.8 37.3

**Table 2 membranes-10-00158-t002:** The ion concentration in the lake and Baltic Sea water used and in the obtained MD retentate (lake water, T_F_ = 353 K).

Ions (mg/L)	Na^+^	Cl^−^	Mg^2+^	Ca^2+^	K^+^	NO_3_^−^	SO_4_^2−^
Lake water	25	47	18	65	7	1.1	88
MD retentate	175 ± 3	338 ± 5	115 ± 4	265 ± 11	46 ± 2	8 ± 1	603 ± 6
Baltic Sea	2329	3720	504	137	78	7.5	538

**Table 3 membranes-10-00158-t003:** The results of mercury porosimetry measurements.

Membrane	Total Pore Area (m^2^)	Median Pore Diameter (μm)	Average Pore Diameter (μm)	Porosity (%)
Accurel PP S6/2	75.3	0.47	0.137	71.2
C-PP	75.1	0.46	0.151	70.7

**Table 4 membranes-10-00158-t004:** The results of XRD analysis—crystallographic parameters.

T_F_ [K]	Mineral Name	ICPDS Code	Chemical Formula	a [Å]	b [Å]	c [Å]
353	Calcite	04-012-0489	CaCO_3_	4.9870	4.9870	17.0580
315	Calcite	04-002-9082	CaCO_3_	4.9674	4.9674	16.9964
315	Calcite magnesian	01-089-1304	(Mg0.03Ca0.97) (CO_3_)	4.9780	4.9780	16.9879
315	Calcite magnesian	01-089-1305	(Mg0.06Ca0.94) (CO_3_)	4.9630	4.9630	16.9570

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
