# Peer review of "Mitigation of Membrane Wetting by Applying a Low Temperature Membrane Distillation"

_membranes, 2020, doi:10.3390/membranes10070158_

Round 1
Reviewer 1 Report
As a conventional desalination approach, MD is studied here at low temperature with the hope to mitigate membrane wetting. The results sound good, while some logical issues, and the novelty of scientific contribution should be improved further.
Membrane selection, is contact angle the sole parameter to consider? Membrane wetting is a complex process with lots of factors to be involved. Beside the contact angle, the membrane structures, pore formation, supporting layer & active layer should be well taken. Otherwise, it is quite far away from the commercial application.
The autopsy of fouled membrane to reveal fouling mechanism or mitigation of membrane wetting is not clear. SEM is only to qualitatively show some typical points of membrane surface, hard to investigate the details at the interface between membrane and feed solutes in a dynamic process.
Is it possible to consider a CFD model to simulate the scaling formation and dissolution process during the MD process at various temperature?
What is the advise or significant impact on real engineering application?
Reviewer 2 Report
I have read the manuscript “Mitigation of membrane wetting by applying a low temperature membrane distillation” by Marek Gryta (MS # membranes-863316) submitted for the publication in Membranes.
The author reports the long term investigations performed on two different hollow fiber reactors in MD processes. It was found that a lower feed temperature reduces scaling and wetting. Three different type of water were used.
The paper is interesting and deserves publication in Membranes after some minor revisions. In particular:
1. The author cited several papers from literature including some personal ones. It should be evidenced the real novelty of this manuscript compared to previous ones (e.g. Fibers 2019, 7(1), Chemical Papers 2019, 73, 591–600, Desalination and Water Treatment 2018, 128, 1–10), where the same membranes were studied with the same waters systems in membrane distillation processes at low temperatures.
2. Table 1 and Table 2: What was the rationale for choosing the two hollow membranes being their parameters identical?
3. Line 288: It is not possible to state that crystals present in Figure 6 are calcite just by SEM. X-Ray or EDX should be use.
4. Line 298: Crystals do not look like porous.
5. Line 311 The symbol “°” should be apex.
6. English should be revised as some sentences sound unconnected.
Reviewer 3 Report
The issue of wetting, surface scaling/fouling of MD membranes are key challenges in the area. The manuscript per se is relevant and points at some of the main aspects requiring attention.
The following could/should be discussed in more depth.
1) Fouling/scaling of MD (hydrophobic membranes) is still a relevant topic, particularly when dealing with complex amphiphilic compounds. Particularly during remediation of process industrial or highly contaminated waters (as the field is moving to right now besides purely desalination) such aspects are critical. The application of MD for such complex solutions was recently demonstrated and could be discussed. Relationships with this study may help further highlight the need and challenges.
https://www.sciencedirect.com/science/article/pii/S0376738819309032
https://www.sciencedirect.com/science/article/pii/S0043135420305479
2) The stability of the MD membranes over time and during operation is also a key challenge in some process waters. This aspect could be further discussed since it seems to be one of the area which the author wants to develop.
3) Surface modification of membranes could also be discussed to highlight the potential to more resilient membranes with more appropriate surface characteristics to prevent scaling.
Reviewer 4 Report
The manuscript concerns an interesting and updated subject. It is well organized and well written. The work can be published in the Journal of Membranes. However, some editing and minor errors should be avoided. Some comments are given as follows:
- The author mentioned that several previous works ([1,7,11,15,24,28-30]) have reported the positive impact of lowering the feed temperature with respect to scaling effect on the membrane behavior. It is important to give further relevant information and main findings of these references. This would help defining the gaps in the literature to be addressed.
- On another side, the author addressed appropriately the effect of lowering the feed temperature on the membrane behavior. However, other parameters of the feed water are also important. For instance the effect of feed velocity can have an important impact on the scaling and fouling phenomena.
- The membrane is submerged in this study. How the obtained results would be used and explored for a conventional MD configuration.
Round 2
Reviewer 1 Report
The manuscript has been well revised, and its present state is ready for publication.